# Study on the Carbon and Nitrogen Isotope Characteristics and Sources and Their Influence on Carbon Sinks in Karst Reservoirs

**Zhongfa Zhou [1,2,3,\*], Jie Kong [1,2,3], Fuqiang Zhang [1,2,3], Yan Zou [1,2,3], Jiangting Xie [1,2,3] and Chaocheng Wen [1,2,3]**

[1] School of Karst Science, Guizhou Normal University, Guiyang 550001, China
[2] School of Geography & Enviromental Science, Guizhou Normal University, Guiyang 550001, China
[3] The State Key Laboratory Incubation Base for Karst Mountain Ecology Environment of Guizhou Province, Guiyang 550001, China
\* Correspondence: fa6897@gznu.edu.cn; Tel.: +86-13985026897

**Abstract:** The hydrochemical analysis method was used to reveal the sources and spatiotemporal variations of carbon and nitrogen elements in the Pingzhai Reservoir, and the C–N coupling cycle and its influence on the karst carbon sink are discussed. The results show the following: (1) The hydrochemical type of the study area is $HCO_3$-Ca. (2) From the river to the reservoir and then to the reservoir outlet, the values of $HCO_3^-$ and $\delta^{13}C_{DIC}$ showed an opposite trend. The values of $NO_3^-$, $\delta^{15}N$-$NO_3^-$, and $\delta^{18}O$-$NO_3^-$ were different in each stage of the river. (3) $HCO_3^-$ mainly comes from the weathering of carbonate rocks and the oxidative decomposition of organic matter. Nitrate mainly comes from chemical fertilizers, soil organic nitrogen, sewage, and livestock manure. (4) The average proportion of $HCO_3^-$ produced by $HNO_3$ dissolving carbonate rock is 8.38%, but this part does not constitute a carbon sink. Compared with rivers, the proportion of $HCO_3^-$ and ($Ca^{2+}$ + $Mg^{2+}$) produced by $HNO_3$ dissolving carbonate rock in reservoir water is relatively large. The input of nitrate not only pollutes the water body with $NO_3^-$ but also changes the carbon source/sink pattern of the water–rock interaction.

**Keywords:** karst carbon sink; carbon and nitrogen sources; dissolved inorganic carbon isotopes; nitrogen and oxygen isotopes; karst area reservoir





## 1. Introduction

Rivers connect terrestrial and marine ecosystems and are important channels for the transfer and transformation of nutrient elements. The transport of particulate and dissolved matter from rivers to the ocean is of great significance for the material cycle of the ecosystem, and it is estimated that the total amount of particulate sediment and dissolved matter transported into the ocean by rivers worldwide is $15.5 \times 10^9$ t/a and $4 \times 10^9$ t/a, respectively [1]. At the same time, to make full use of water energy resources, water conservation has been pursued around the world in recent decades. The construction of dams transforms a single river ecosystem into a river–reservoir ecosystem, river continuity is forced to change, and the pattern of material transport from the source to the estuary also changes. Reservoirs are usually characterized by poor flow, water temperature stratification, and large depth, and the retention and transformation of various substances in reservoirs are relatively considerable. Many studies have also been carried out on the biogeochemical cycle of river–reservoir ecosystems [2–5].

As essential elements for the growth of life, carbon, and nitrogen play an important role in ecosystems. Related studies have shown that carbon and nitrogen can reflect the construction of aquatic food chains and the division of trophic levels in aquatic ecosystems [6] and can also be used for carbon and nitrogen isotope analysis in aquatic plants and plankton [7,8]. Water bodies in karst areas are generally rich in calcium and somewhat

alkaline. In terms of the impact of nutrients on water quality, even if the concentrations of nitrogen and phosphorus are low, aquatic plants (blue-green algae) and plankton can develop through the carbon fertilization effect. The input of exogenous nitrogen will contribute to the growth of living organisms such as microorganisms and plants in water, which will affect the absorption and decomposition of carbon [3,4].

However, increasing human activities such as land-use change, fossil fuel burning, and agricultural production have altered the natural state of the carbon and nitrogen cycle [9,10]. For example, the use of nitrogen fertilizer in agriculture can promote photosynthesis by increasing the net primary productivity of vegetation and microbial activities and increase the decomposition of organic matter by microorganisms, but this will produce more $CO_2$, part of which is re-discharged into the atmosphere as a carbon source, and the other part infiltrates into the karst surface zone and vadose zone with rainfall to form carbonic acid, increasing the dissolution of carbonate rocks [11,12]. Relevant studies also show that nitric acid produced by agricultural and urban activities interferes with the karst carbon cycle [13–15]. In contrast to carbonate rocks dissolved by carbonic acid, carbonate rocks dissolved by nitric acid do not consume $CO_2$ in the atmosphere/soil, leading to an increase in $HCO_3^-$ and ($Ca^{2+}$ + $Mg^{2+}$) concentrations in water, playing the opposite role in $CO_2$ emission reduction [16,17]. Baker et al. (2008) showed that a river flowing through the city had the highest dissolved inorganic carbon (DIC) concentration in the carbonate rock area of Britain [18]. Barnes et al. (2009) found that DIC was higher in watersheds dominated by urban land than in watersheds dominated by forestland [13]. It is estimated through laboratory simulation that fertilization of cultivated land in karst regions will lead to an additional increase in the ($Ca^{2+}$ + $Mg^{2+}$) concentration in rivers by $5.7 \times 10^{12}$ mmol/a and will release $CO_2$ to the atmosphere [19]. A study in a karst basin in southwest France found that the application of chemical fertilizers increased the concentration of nitric acid in river water and estimated that the amount of atmospheric $CO_2$ absorbed by the weathering of carbonate rocks in the basin decreased by 7–11%, and the karst carbon sink decreased by 5.7–13.4% in the whole region of France [20]. According to Brunet et al. (2011), the nitric acid formed by nitrification of nitrogen fertilizer can cause soil and water acidification, increase the concentration of alkaline cations, change the carbon budget, and actively participate in the weathering of carbonate rocks [21]. A study in the typical karst agricultural area of Southwest China shows that $H^+$ released from the nitrification of nitrogen fertilizer accelerates the weathering of carbonate rocks, which not only reduces the consumption of atmospheric $CO_2$ but also increases the $HCO_3^-$ flux by approximately 20% [22].

Southwest China has the largest contiguous distribution of carbonate rocks in the world, with an exposed area of carbonate rocks of $54 \times 10^4$ km² [23]. Southwest China has also become a key area for studying carbon and nitrogen cycling. The Pingzhai Reservoir is located in Southwest China, and its water transfer scope involves the Yangtze River basin and the Pearl River basin [24]. Its carbon and nitrogen concentrations are of great significance for the water quality security of the Yangtze and Pearl Rivers. In terms of geological background, the Pingzhai Reservoir is located in a deep river canyon, with very thick carbonate rock strata distributed on both sides of the canyon. Affected by karstification, a multilayered karst hydrogeological structure has been formed. The karst morphology mainly includes peak-cluster depressions, dissolving gullies and troughs, falling caves, funnels, karst caves, and karst pipeline systems. In the region, in recent years, with the increase in population, frequent industrial and agricultural activities in the basin, and the large amount of agricultural fertilizers used with low utilization efficiency [24,25], nutrients have entered rivers and reservoirs through cracks in the karst and underground rivers along with the runoff and pore water generated by precipitation, leading to the accumulation of nitrogen nutrients and participating in water-rock interactions. Previous studies on carbon and nitrogen in the Pingzhai Reservoir and its inflow river were relatively isolated [25–27], but with the progress of research, it has been found that it is very important to explore the coupling of carbon and nitrogen and its environmental effects. The objectives of this study were to explore the water hydrochemical types and the spatial and temporal

distribution characteristics of nitrate nitrogen and oxygen isotopes and dissolved inorganic carbon isotopes in a karst reservoir basin, determine the sources of DIC and nitrate in water, explore the carbon and nitrogen coupling cycle in different periods, and quantitatively evaluate the impact of the carbon and nitrogen coupling cycle and nitric acid from external sources on carbonate dissolution in the basin. This study can provide a reference for the study of the effect of C–N coupling and karst carbon sinks on the river–reservoir continuum in karst regions.

## 2. Materials and Methods

### 2.1. Overview of the Study Area

The Pingzhai Reservoir (105°17'3″ E–105°26'44″ E, 26°29'33″ N–26°35'38″ N) is the source reservoir of Guizhou's Central Water Control Project and undertakes the functions of irrigation, drinking water supply, and power generation in the region (Figure 1). The reservoir is formed by the convergence of five rivers (Nayong River, Shuigong River, Zhangwei River, Baishui River, and Hujia River) in the upper reaches of the reservoir, and the drainage area is 833.77 km$^2$. The construction of the reservoir was completed in 2015, the maximum dam height is 157.5 m, the maximum water level is 1331 m, the regulated storage capacity is 448 million m$^3$, and the total storage capacity is 1.089 billion m$^3$. The study area is located in the subtropical monsoon climate zone, summer is hot and rainy, winter is mild and slightly rainy, the annual average temperature is 14 °C, the annual average rainfall is between 1200 and 1500 mm, and the rainfall has seasonal differences under the influence of the monsoon climate. The wet season is from May to August, the dry season is from November to February, and the normal season is March, April, September, and October.

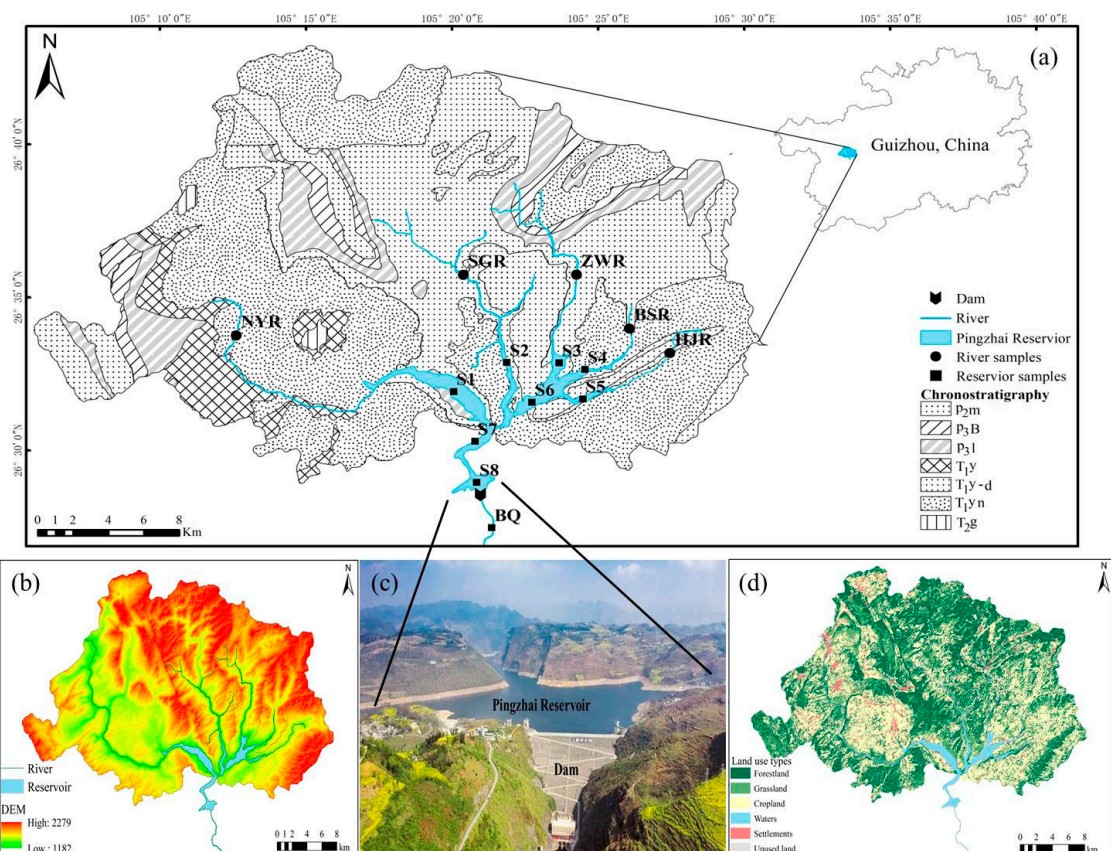

**Figure 1.** Overview map of the study area ((**a**) is the chronostratigraphic diagram and sampling points distribution of the study area; (**b**) is the DEM of the study area; (**c**) is the reservoir and dam; (**d**) is the land use types map).

The terrain and geomorphology of the study area are complex and belong to the middle-low mountain valley landform of tectonic dissolution and erosion. The geological structure pattern is controlled by the Yanshan movement, and anticlines, synclines, and compressive faults are formed. As shown in Figure 1, the main outcrops are the Permian Dalong Formation ($P_3d$), Longtan Formation ($P_3l$), Maokou Formation ($P_2m$), Triassic Yongningzhen Formation ($T_1yn$), Yelang Formation ($T_1y$), and Guanling Formation ($T_2g$). The stratigraphic lithology is carbonate rocks such as limestone and dolomite and clastic rocks such as sandy mudstone, shale intercalated marl, and coal. The main land-use types in the study area were cultivated land and forestland, followed by grassland, construction land, and unused land. Affected by regional lithology and crop cultivation habits, the main soil types are yellow-brown soil, lime soil, and yellow soil, in addition to a small amount of paddy soil. The main crops are rice, corn, and potato. The main fertilizers used in the region are ammonium-based N fertilizer (urea), nitrogen–phosphorus compound fertilizer, and animal manure; the applied pesticides mainly include insecticides and rust removers, which pose the risk of agricultural nonpoint source pollution.

## 2.2. Sample Collection and Analysis

In accordance with the sampling conditions of the study area and Technical Provisions for the Design of Water Quality Sampling Schemes (HJ495-2009), sampling points NYR, SGR, ZWR, BSR, and HJR were set up in the upper reaches of the five rivers, and sampling points S1, S2, S3, S4, and S5 were set up at the intersection of the rivers and reservoir areas. Three sampling points (S6, S7, and S8) were set up in the reservoir area, and one sampling point was set up at the outlet of the dam (BQ) (Figure 1), for a total of 14 sampling points. Water samples were collected from the study area in November 2020 and January and July 2021, representing the normal, dry, and wet periods, respectively. Water pH, water temperature (WT), electrical conductivity (EC), and dissolved oxygen (DO) were measured in the field by a WTW Multi3430 portable multiparameter water quality analyzer with an accuracy of 0.001 pH unit, 0.01 °C, 1 μs/cm, and 0.01 mg/L, respectively. The concentrations of $HCO_3^-$ and $Ca^{2+}$ were titrated onsite using an alkalinity kit and calcium kit (Merck, Germany) with accuracies of 0.1 mmol/L and 2 mg/L, respectively. The collected water samples were filtered through a 0.45-μm filter membrane and loaded into polyethylene sampling bottles that had been precleaned with deionized water. Concentrated nitric acid was added to the water samples to pH < 2 for determination of cation concentration, and 2 drops of $HgCl_2$ were added to the water sample to inhibit microbial activity for determination of dissolved inorganic carbon isotope ($\delta^{13}C_{DIC}$).

Anions, cations, and $\delta^{13}C_{DIC}$ were measured at the State Key Laboratory of Environmental Geochemistry, Institute of Geochemistry, Chinese Academy of Sciences, and $\delta^{15}N$-$NO_3^-$ and $\delta^{18}O$-$NO_3^-$ were measured at the Analysis and Testing Center of the Third Institute of Oceanography, Ministry of Natural Resources. The concentration of cations ($K^+$, $Na^+$, $Mg^{2+}$) was determined by Inductively Coupled Plasma-Emission Spectrometer (VISTA MPX, Varian, USA), and the concentration of anions ($NO_3^-$, $Cl^-$, $SO_4^{2-}$) was determined by ion chromatography (ICS90, Dionex, Sunnyvale, CA, USA). The limit of detection was 0.01 mmol/L. The method for the determination of water body $\delta^{13}C_{DIC}$ was to add 100% pure phosphoric acid into the injection bottle (vacuumized) and injection high purity helium gas, then inject 20 mL water sample into the injection bottle with a syringe, and heat it in a 60 °C water bath beaker. The $CO_2$ produced by the reaction was separated by a cold trap and then loaded with helium into a Finnigan MAT253 gas isotope mass spectrometer for determination. The bacterial denitrification method was used for the determination of $\delta^{15}N$-$NO_3^-$ and $\delta^{18}O$-$NO_3^-$. Denitrifying bacteria (ATCC 13985, DSM 6698) without nitrous oxide reductase activity were used to terminate the reaction after reducing $NO_3^-$ to $N_2O$, thus obtaining nitrogen and oxygen in $N_2O$ from the $NO_3^-$ in the sample [28,29]. A GasBench continuous flow gas introduction instrument and MAT 253 stable isotope ratio mass spectrometer were used to determine the $\delta^{15}N$ and $\delta^{18}O$ contents in $N_2O$. To ensure the accuracy of the obtained measurements, reference materials USGS34

($\delta^{15}$N = −1.8%, $\delta^{18}$O = −27.9%), USGS32 ($\delta^{15}$N = +180%, $\delta^{18}$O = +25.7%), and IAEA-NO$_3$ were used. The test accuracy of $\delta^{13}$C$_{DIC}$ was 0.2%, and the result is reported as parts pel mil (%) relative to the Vienna PDB reference standard. The test accuracy of $\delta^{15}$N-NO$_3^-$ and $\delta^{18}$O-NO$_3^-$ was 0.3%. Atmospheric nitrogen (N$_2$) and Vienna standard mean seawater (V-SMOW) were used as references for the $\delta^{15}$N and $\delta^{18}$O results, respectively.

### 2.3. Flux Calculation

HCO$_3^-$ and NO$_3^-$ flux was calculated from the total water flow multiplied by the concentration of HCO$_3^-$ and NO$_3^-$ [30]. Flux is calculated using the equation:

$$flux_{C/N} = con_{C/N} \times Q \tag{1}$$

where $flux_{C/N}$ refers to annual HCO$_3^-$ and NO$_3^-$ flux (t·a$^{-1}$), $con_{C/N}$ is the concentration of HCO$_3^-$ and NO$_3^-$ (mg/L), and $Q$ refers to the water discharge in unit time (m$^3$·a$^{-1}$).

## 3. Results

### 3.1. Physicochemical Indices and Hydrochemical Characteristics of Water

The physicochemical indices of the water body of Pingzhai Reservoir and its inflow river showed seasonal changes (Table 1). The water temperature ranged from 9.15 °C to 26.65 °C, with an average temperature of 16.92 °C. The pH value of water ranged from 7.89 to 10.67, with an average value of 8.69, generally showing the characteristics of weakly alkaline water. The EC of the water body varied greatly (196–578 μs/cm) and showed the temporal pattern dry season > normal season > wet season. In terms of the water DO concentration, the annual variation ranged from 6.39 to 11.51 mg/L, with an average of 8.5 mg/L. The water body was generally in an aerobic state, which was conducive to the occurrence of nitrification, and temporally, DO was the highest in the wet season and the lowest in the dry season.

**Table 1.** Main hydrochemical parameters of water bodies in different seasons.

| Index | Normal Season | | | Dry Season | | | Wet Season | | |
|---|---|---|---|---|---|---|---|---|---|
| | River | Reservoir | Dam | River | Reservoir | Dam | River | Reservoir | Dam |
| WT (°C) | 15.3 ± 0.9 | 17 ± 0.3 | 16.9 | 11.3 ± 0.9 | 11.4 ± 0.7 | 12.9 | 19.7 ± 0.3 | 25.1 ± 0.7 | 21.7 |
| pH | 8.3 ± 0.2 | 8.8 ± 0.1 | 8.1 | 8.4 ± 0.3 | 8.5 ± 0.1 | 8.6 | 9.6 ± 0.9 | 8.9 ± 0.1 | 7.9 |
| EC (μs/cm) | 392.6 ± 2.1 | 328.9 ± 1.5 | 390 | 335.2 ± 2.1 | 421.1 ± 1.5 | 434 | 316.6 ± 0.7 | 312.6 ± 0.6 | 421 |
| DO (mg/L) | 8.4 ± 0.2 | 8.7 ± 0.3 | 7.3 | 9.2 ± 0.5 | 6.7 ± 0.2 | 8.5 | 7.8 ± 0.2 | 10.4 ± 0.8 | 7.8 |
| Ca$^{2+}$ (mg/L) | 48.6 ± 0.9 | 49.1 ± 0.6 | 55.2 | 61.1 ± 0.9 | 61.3 ± 0.8 | 69.9 | 52.1 ± 0.5 | 36.8 ± 0.3 | 78.3 |
| Na$^+$ (mg/L) | 23.9 ± 0.8 | 10.7 ± 0.3 | 12.1 | 34.4 ± 1.3 | 13.9 ± 0.3 | 19.6 | 5.6 ± 0.5 | 13.6 ± 0.1 | 3.3 |
| Mg$^{2+}$ (mg/L) | 5.2 ± 0.9 | 5.7 ± 0.7 | 6.9 | 6.2 ± 0.5 | 6.6 ± 0.3 | 7.4 | 4.1 ± 0.4 | 6.9 ± 0.1 | 6.4 |
| K$^+$ (mg/L) | 2.2 ± 0.6 | 2.0 ± 0.2 | 2.4 | 2.1 ± 0.8 | 2.2 ± 0.7 | 2.1 | 3.2 ± 0.6 | 3.4 ± 0.4 | 3.0 |
| HCO$^{3-}$ (mg/L) | 165.3 ± 3.3 | 133.8 ± 0.8 | 186.1 | 169.6 ± 1.7 | 160.1 ± 0.6 | 164.7 | 126.3 ± 1.3 | 91.5 ± 0.9 | 192.1 |
| NO$^{3-}$ (mg/L) | 9.6 ± 0.8 | 11.6 ± 0.7 | 13.6 | 10.6 ± 0.4 | 12.6 ± 0.1 | 15.2 | 12.4 ± 0.8 | 9.8 ± 0.7 | 17.5 |
| Cl$^-$ (mg/L) | 5.4 ± 0.7 | 4.5 ± 0.5 | 5.8 | 8.9 ± 0.9 | 6.1 ± 0.5 | 10.9 | 3.9 ± 1.1 | 9.1 ± 0.9 | 5.6 |
| SO$_4^{2-}$ (mg/L) | 70.5 ± 0.8 | 49.5 ± 0.3 | 60.8 | 75.9 ± 0.7 | 59.1 ± 0.3 | 70.6 | 39.6 ± 1.8 | 57.9 ± 0.9 | 27.7 |
| $\delta^{13}$C$_{DIC}$ (%) | −12.3 ± 0.8 | −9.8 ± 0.6 | −11.5 | −10.9 ± 0.5 | −9.9 ± 0.3 | −12.7 | −12.7 ± 0.6 | −5.3 ± 0.4 | −13.0 |
| $\delta^{15}$N-NO$_3^-$ (%) | 2.2 ± 0.5 | 1.4 ± 0.7 | 2.3 | 14.4 ± 0.7 | 15.9 ± 0.6 | 14.0 | 5.4 ± 0.8 | 7.3 ± 0.6 | 4.3 |
| $\delta^{18}$O-NO$_3^-$ (%) | 5.1 ± 0.6 | 3.0 ± 0.9 | 4.1 | 0.9 ± 0.7 | 2.5 ± 0.1 | 0.5 | 20.8 ± 0.3 | 21.8 ± 0.2 | 22.0 |

Note: mean ± standard deviation (SD).

The total cationic equivalent concentration (TZ$^+$ = 2Ca$^{2+}$ + 2Mg$^{2+}$ + K$^+$ + Na$^+$) in the Pingzhai Reservoir and its inflow rivers ranged from 1.92 to 14.40 meq/L, with an average value of 8.16 meq/L. The total anion equivalent concentration (TZ$^-$ = HCO$_3^-$ + NO$_3^-$ + Cl$^-$ + 2SO$_4^{2-}$) ranged from 1.97 to 10.24 meq/L, with an average value of 6.10 meq/L. Taking the river and reservoir area together, the total cationic equivalent concentration of river water ranged from 3.41 to 14.39 meq/L, with an average of 8.91 meq/L. The total anion equivalent concentration ranged from 2.37 to 10.57 meq/L, with an average of 6.47 meq/L. The total equivalent concentrations of cations and anions in the reservoir were 1.97–9.45 meq/L and 3.24–6.01 meq/L, with averages of 5.71 meq/L and 4.63 meq/L, respectively. The Piper diagram can directly reflect the composition characteristics of the main ions in water (Figure 2). The predominant cations in the Pingzhai

Reservoir were $Ca^{2+}$ and $Na^+$, whose contents accounted for 68% and 21% of the total cations, respectively. The dominant anions were $HCO_3^-$ and $SO_4^{2-}$, which accounted for 65% and 27% of the total anions, respectively. According to the Shukalev classification, the hydrochemical type in the study area was the $HCO_3$-Ca type.

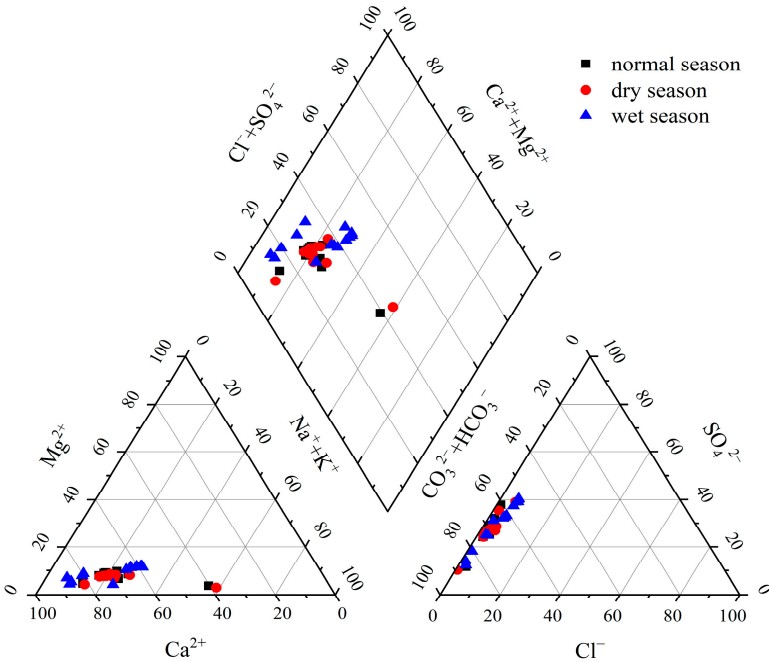

**Figure 2.** Piper diagram of hydrochemical types in the Pingzhai Reservoir basin.

### 3.2. Characteristics of DIC and $\delta^{13}C_{DIC}$ in the Pingzhai Reservoir

Influenced by the dissolution of carbonate rocks, the water in the study area is weakly alkaline, under which condition $HCO_3^-$ is the main form of DIC [31]. In this study, $HCO_3^-$ is used to represent the concentration of DIC in the water. The variation in $HCO_3^-$ concentration in the Pingzhai Reservoir and inflow rivers ranged from 1.30 to 3.45 mmol/L, with an average of 2.32 mmol/L. The average concentrations of $HCO_3^-$ in river, reservoir, and outlet water were 2.52 mmol/L, 2.11 mmol/L, and 2.97 mmol/L, respectively. The seasonal variation was as follows: dry season (2.69 mmol/L) > normal season (2.44 mmol/L) > wet season (1.82 mmol/L). The average annual flows of the NYR, SGR, and ZWR were $1.35 \times 10^9$ m$^3$·a$^{-1}$, $5.89 \times 10^8$ m$^3$·a$^{-1}$, and $5.22 \times 10^8$ m$^3$·a$^{-1}$, respectively. Flow data from BSR and HJR are lacking. According to Equation (1), combined with the $HCO_3^-$ concentration at the monitoring section, it can be calculated that the $HCO_3^-$ fluxes of NYR, SGR, and ZWR were 2580.25 t·a$^{-1}$, 700.62 t·a$^{-1}$, and 722.52 t·a$^{-1}$, respectively. The $\delta^{13}C_{DIC}$ value ranged from −3.1 to −18.4‰, with an average value of −9.9‰, and the average $\delta^{13}C_{DIC}$ values in river, reservoir, and outlet water were −12.0‰, −8.3‰, and −12.4‰, respectively. In terms of time, the $\delta^{13}C_{DIC}$ value was negative in the wet season and positive in the dry season. There was no correlation between $HCO_3^-$ and $\delta^{13}C_{DIC}$ in the normal and dry periods except for a negative correlation between $HCO_3^-$ and $\delta^{13}C_{DIC}$ in the wet season (r = −0.692, $p < 0.01$). This is due to the strong photosynthesis during the wet season. Phytoplankton absorb the DIC in water and fractionate $\delta^{13}C_{DIC}$, resulting in a decrease in $HCO_3^-$ concentration and an increase in the $\delta^{13}C_{DIC}$ value. Due to the influence of the dam, stable stratification is formed in the water body due to temperature differences in summer, which blocks the material exchange between the surface water body and the deep-water body. However, with the disappearance of stable stratification of water temperature in the normal and dry periods, the high concentration of $HCO_3^-$ at the bottom diffuses to the surface water with water turnover and internal circulation, resulting in a high concentration of $HCO_3^-$ in the surface water.

*3.3. Characteristics of $NO_3^-$, $\delta^{15}N$-$NO_3^-$, and $\delta^{18}O$-$NO_3^-$ in the Pingzhai Reservoir*

The variation in the $NO_3^-$ concentration in the Pingzhai Reservoir ranged from 1.92 to 17.47 mg/L, with an average of 11.46 mg/L, and the time variation was wet season (12.68 mg/L) > dry season (12.09 mg/L) > normal season (11.02 mg/L). The average $NO_3^-$ concentrations in the river, reservoir area, and outlet were 11.33 mg/L, 10.68 mg/L, and 15.4 mg/L, respectively. According to the average annual flow and concentration of $NO_3^-$, the $NO_3^-$ fluxes of the NYR, SGR, and ZWR points were calculated as 210.74 t·a$^{-1}$, 25.33 t·a$^{-1}$, and 54.76 t·a$^{-1}$, respectively. The $\delta^{15}N$-$NO_3^-$ values in the dry season ranged from 13.0 to 17.1%, and the $\delta^{15}N$-$NO_3^-$ values in the wet and normal seasons ranged from 3.4 to 8.3% and 0.4 to 2.9%, respectively. This seasonal variation indicates different nitrate sources. The $\delta^{18}O$-$NO_3^-$ value was the highest in the wet season, intermediate in the normal season, and the lowest in the dry season.

## 4. Discussion

*4.1. Spatial Variation and Influencing Factors of Dissolved Inorganic Carbon and Nitrate*

Subject to different environmental factors, carbon and nitrogen elements and isotopes will show different spatial and temporal changes (Figure 3). Compared with the reservoir area, the $\delta^{13}C_{DIC}$ value of the river is more negative, and the DIC concentration is higher, which is due to the large proportion of farmland and woodland in the river flow area. The proportion of cultivated land area in the Baishui and Hujia River Basins is more than 50%, and the proportion of woodland area in the Shuigong River Basin is nearly half [24]. The $\delta^{13}C_{DIC}$ value of the water body inherits the $\delta^{13}C_{DIC}$ value of vegetation and soil $CO_2$. The amount of soil $CO_2$ produced by plant root respiration into the water body increased, which led to a negative $\delta^{13}C_{DIC}$ value and an increase in DIC concentration. In addition, the Narong and Zhangwei Rivers flow through cities and towns, and human activities (industrial production and agricultural cultivation) discharge sewage with negative $\delta^{13}C_{DIC}$ values into the river, which will also cause a negative $\delta^{13}C_{DIC}$ value in river water [32].

Compared with the reservoir area, the $\delta^{13}C_{DIC}$ value at the outlet of the reservoir was negative, and the DIC concentration was higher. The spatial variation in $\delta^{13}C_{DIC}$ values was negative in the river, positive in the reservoir area, and negative in the outlet. The DIC concentration first increased, then decreased, and then increased again. Yuan et al. (2021) studied DIC concentration and $\delta^{13}C_{DIC}$ in cascade reservoirs of the Yunnan section of the Lancang River and found that the DIC concentration was high in the river, low in the reservoir, and high at the outlet during the wet and dry seasons, and the $\delta^{13}C_{DIC}$ value first showed an increasing and then decreasing trend [4]. In the karst area of Southwest China, the DIC concentration of the Hongjiadu Reservoir in the Wujiang River Basin is higher at the outlet than in the river and reservoir area, and the $\delta^{13}C_{DIC}$ value is negative in the river, positive in the reservoir area, and negative at the outlet. The DIC concentration in the rainy season is lower than that in the dry season, and the $\delta^{13}C_{DIC}$ value in the rainy season is higher than that in the dry season [33]. All of these parameters showed the same change trend in the Pingzhai Reservoir basin, which was due to the following: Under the influence of the subtropical monsoon climate, the rainfall in the study area is mostly concentrated in spring and summer, the increase in rainfall leads to an increase in runoff into the reservoir, and the DIC concentration decreases due to the influence of water dilution. Generally, photosynthesis on the water surface is relatively active, and aquatic phytoplankton will produce approximately 18–20% isotope fractionation while absorbing $CO_2$ through photosynthesis [34]. Therefore, the DIC concentration of surface water in the reservoir area is low, and the $\delta^{13}C_{DIC}$ value is positive, while the photosynthesis of the bottom water is weak, and the degradation of organic matter at the water–sediment interface produces $CO_2$ with poor $^{13}C$, which increases the DIC concentration and causes the $\delta^{13}C_{DIC}$ value to be negative. As a result, the DIC value of the lower water is higher than that of the upper water, and the $\delta^{13}C_{DIC}$ value of the lower water is lower than that of the upper water. The discharge mode of the Pingzhai Reservoir is bottom discharge, so bottom water with high DIC concentrations and low $\delta^{13}C_{DIC}$ values is injected downstream.

There are also relevant studies that show that the DIC concentration in the water body is lower in the rainy season than in the dry season, and the $\delta^{13}C_{DIC}$ value in the rainy season is also lower than that in the dry season, which may be related to the water nutrient level, reservoir operation mode, vegetation cover conditions, and tributary inflow [4,35,36].

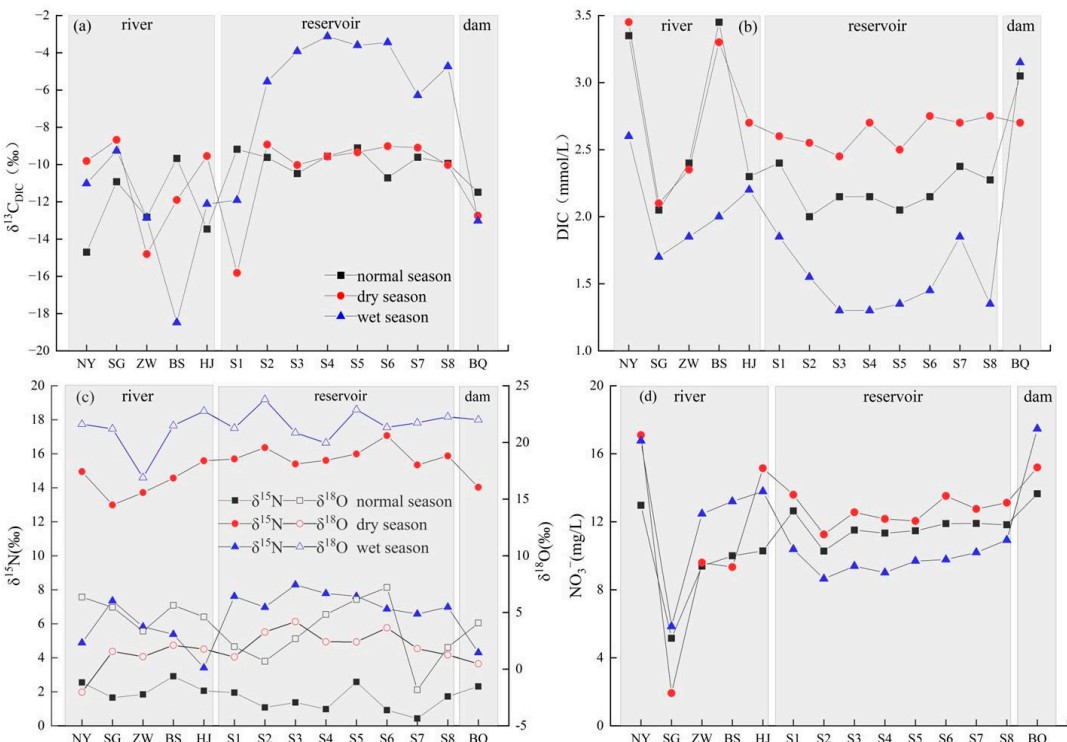

**Figure 3.** Variations of $\delta^{13}C_{DIC}$ and DIC, $\delta^{15}N$-$NO_3^-$, $\delta^{18}O$-$NO_3^-$, and $NO_3^-$ along the course of water bodies in different seasons ((**a**) is the variation characteristic of $\delta^{13}C_{DIC}$; (**b**) is the variation characteristic of DIC; (**c**) is the variation characteristic of $\delta^{15}N$, $\delta^{18}O$; (**d**) is the variation characteristic of $NO_3^-$).

The concentration of $NO_3^-$ has a more direct impact on the deterioration of water quality. In the river section, the concentration of nitrate in the Nayong and Hujia Rivers was higher, which is a consequence of the discharge of domestic sewage and livestock breeding wastewater increasing the concentration of nitrate in the water body. The Shuigong River sampling site, which was not surrounded by residential and cultivated land, was minimally affected by anthropogenic activities and therefore had the lowest nitrate concentration. In the reservoir area, the water was fully mixed, and the nitrate concentration differed little among the sampling points in different seasons. Similar to the distribution of the $NO_3^-$ concentration, the $\delta^{15}N$-$NO_3^-$ and $\delta^{18}O$-$NO_3^-$ values also showed large differences in the river section and small differences in the reservoir section. It is mainly affected by crop cultivation and fertilization, domestic sewage discharge, and other factors. Fadhullah et al. (2020) used nitrate nitrogen and oxygen isotopes to study the source of nitrate in the Bukit Merah Reservoir in Southeast Asia and found that industrial and mining production and agricultural expansion in the upstream river disturbed the value of nitrate nitrogen and oxygen isotopes in the river water. The nitrogen and oxygen isotopes in the reservoir area indicated that the nitrate concentration is affected by atmospheric deposition [37]. Studies on nitrate isotopes in Chaohu Lake and its rivers showed that nitrification of soil organic nitrogen and soil erosion caused changes in nitrate isotopes [38].

### 4.2. Sources of Dissolved Inorganic Carbon and Nitrate

The DIC concentration and $\delta^{13}C_{DIC}$ value in water record and reflect the geochemical behavior and cycling characteristics of carbon. Because carbon is affected by many factors

in the cycling process and different sources of carbon have different isotopic values, the $\delta^{13}C_{DIC}$ value can be used to trace the source of DIC. Related studies have shown that DIC in natural water mainly comes from atmospheric $CO_2$ (including $CO_2$ in atmospheric precipitation), dissolution of carbonate rock, and biogenic $CO_2$ (plant root respiration and organic matter decomposition) dissolution in water [38,39]. The study area is located in the acid rain region of Southwest China, the pH value of rainwater is weakly acidic, the DIC concentration is low [40], and the partial pressure of water $CO_2$ ($pCO_2$) in the region is higher than that of atmospheric $CO_2$ [23,35]. The influence of atmospheric precipitation and atmospheric $CO_2$ on DIC and $\delta^{13}C_{DIC}$ values is not considerable. Therefore, the DIC in water is mainly derived from the weathering of carbonate rocks and the oxidative decomposition of organic matter.

The sources of nitrate in aquatic ecosystems mainly include atmospheric deposition, soil organic nitrogen, chemical fertilizer, domestic sewage, and livestock manure. Atmospheric deposition of $\delta^{15}N\text{-}NO_3^-$ ranges from $-8\%$ to $+15\%$, and of $\delta^{18}O\text{-}NO_3^-$ ranges from $+60\%$ to $+95\%$. The nitrogen and oxygen isotope values of nitrate fertilizer range from $-5\%$ to $+5\%$ and $+17\%$ to $+25\%$, respectively. The nitrogen and oxygen isotopes of nitrate produced by fertilizer and deposition of $NH_4^+$, soil organic nitrogen, sewage, and livestock manure range from $-10\%$ to $+25\%$ and $-10\%$ to $+10\%$, respectively [41–44]. Combined with the $\delta^{15}N\text{-}NO_3^-$ and $\delta^{18}O\text{-}NO_3^-$ values of the water body in the study area, various nitrate sources could be determined (Figure 4). During the normal season, $\delta^{15}N\text{-}NO_3^-$ and $\delta^{18}O\text{-}NO_3^-$ were distributed in soil organic nitrogen and ammonium fertilizer end-members, indicating that nitrification of soil organic nitrogen and ammonium fertilizer was the main source of nitrate in water. In the dry season, $\delta^{15}N\text{-}NO_3^-$ and $\delta^{18}O\text{-}NO_3^-$ are distributed in domestic sewage endmembers and livestock manure, and $\delta^{15}N\text{-}NO_3^-$ is higher. Ren et al. (2021) found that the $\delta^{15}N$ value of some points was higher in a study of groundwater in the Zhaotong Basin, Yunnan Province [8]. Lin et al. (2019) also found a similar phenomenon in a study of the Illinois River in Chicago, USA [45]. This is because the volatilization of $NH_4^+$ in sewage releases a large amount of lighter isotopes, making the remaining $NO_3^-$ enriched in heavy $\delta^{15}N$. The $\delta^{15}N\text{-}NO_3^-$ and $\delta^{18}O\text{-}NO_3^-$ were distributed in the nitrate fertilizer endmembers during the wet season. It may be that inorganic nitrogen from the oxidation and decomposition of nitrate fertilizer enters the water with surface runoff. In general, agricultural activities (ammonium fertilizer, nitrate fertilizer, soil organic nitrogen, livestock manure, and sewage) were important sources of nitrate in the Pingzhai Reservoir. Chemical fertilizers (ammonium fertilizer, nitrate fertilizer) can improve soil fertility and increase crop yield. From 1997 to 2005, the amount of chemical fertilizer applied in China increased from 7.07 million tons to 26.21 million tons [46]. However, the irrational use of chemical fertilizers and the lack of farmland management often lead to nitrogen loss and nitrate pollution. Due to the periodic storage and discharge of the reservoir, a water-level fluctuation zone will form around the reservoir, and the water-level fluctuation zone is also the area where agricultural nonpoint source pollution and soil erosion often occur [47]. According to Han et al. (2016), the amount of soil erosion in the Yangtze River basin and Wujiang River basin reached $1.4 \times 10^{10}$ t/a and $1.2 \times 10^9$ t/a in 2014, respectively [48]. Soil erosion in the water-level fluctuation zone will transport a large amount of nutrients and soil organic nitrogen to rivers and reservoirs. This part of the lost N will be converted into nitric acid through nitrification and will participate in the geochemical cycle, which will affect the chemical composition of rivers and reservoirs, especially the concentrations of $Ca^{2+}$ and $HCO_3^-$ [16]. Nitric acid participates in the dissolution of carbonate rocks and directly or indirectly releases $CO_2$ into the atmosphere, and these anthropogenic carbon source emissions offset part of the natural carbon sink in the natural process [20]. Therefore, it is necessary to consider the impact of external acid input on the weathering of carbonate rocks in the basin.

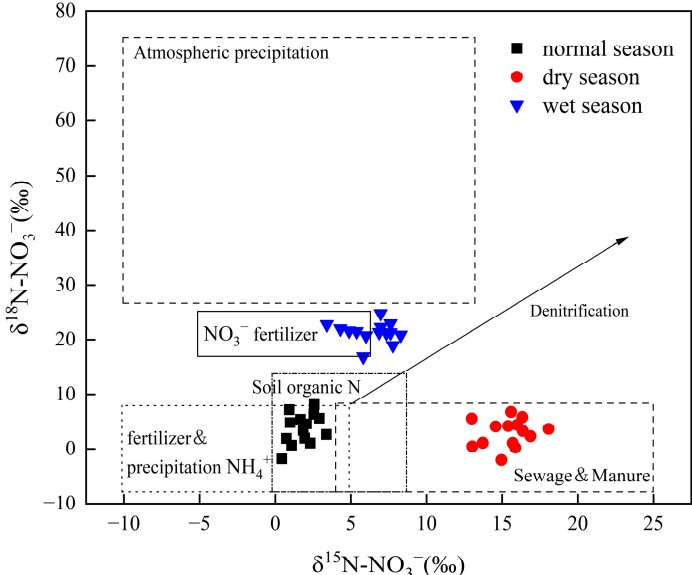

**Figure 4.** Distribution characteristics of $\delta^{15}N$-$NO_3^-$ and $\delta^{18}O$-$NO_3^-$ in water. The isotopic values of various nitrate sources are based on Kendall et al. (2007) [44].

*4.3. Weathering of Carbonate Rocks and C–N Coupling Relationship in the Pingzhai Reservoir Basin*

The average $\delta^{13}C_{DIC}$ of $CO_2$ produced by root respiration of $C_3$ vegetation is −27% because the migration and diffusion of $CO_2$ in the soil layer will produce approximately 4% isotope fractionation, so the $\delta^{13}C_{DIC}$ value of soil $CO_2$ produced by plant respiration and organic matter oxidation decomposition is approximately −23%, and the $\delta^{13}C_{DIC}$ value of marine carbonate rocks in karst areas is 0% [49–51]. According to the stoichiometric relationships of carbonate rock dissolution, with the participation of carbonic acid ($H_2CO_3$), 2 mol $HCO_3^-$ will be produced in the process of carbonate rock dissolution, of which 1 mol is derived from atmospheric/soil $CO_2$ in the watershed and 1 mol from carbonate rocks (Equation (2)) [52].

$$Ca_{(1-x)}Mg_xCO_3 + CO_2 + H_2O = (1-x)Ca^{2+} + xMg^{2+} + 2HCO_3^- \tag{2}$$

In the process of carbonate rock dissolution by carbonic acid, carbon isotopes will produce +9% fractionation, so the water body $\delta^{13}C_{DIC}$ theoretical value is approximately −14% [53]. The variation in $\delta^{13}C_{DIC}$ in the Pingzhai Reservoir water ranges from −3.13% to −18.42%, with an average of −9.92%. Compared with the theoretical value, the $\delta^{13}C_{DIC}$ actual value of the water body is more positive, which indicates that in addition to carbonic acid, there are other sources of acid, such as nitric acid from human activities, which contribute to the dissolution of carbonate rocks in the basin (Equation (3)). In addition, the $[Ca^{2+} + Mg^{2+}]/[HCO_3^-]$ equivalent ratios in the water samples in the study area were all greater than 1 (1.37 on average), and the results indicate that $Ca^{2+}$ and $Mg^{2+}$ in the water sample are surplus relative to $HCO_3^-$, that is, $Ca^{2+}$ and $Mg^{2+}$ have additional sources [54]. In addition to the carbonate rocks dissolved by $H_2CO_3$ and $HNO_3$, the weathering and dissolution of silicate rocks in the watershed can also produce $Ca^{2+}$, $Mg^{2+}$, and $HCO_3^-$ ions (Equations (4) and (5)).

$$Ca_{(1-x)}Mg_xCO_3 + HNO_3 = (1-x)Ca^{2+} + xMg^{2+} + NO_3^- + HCO_3^- \tag{3}$$

$$Ca_xMg_{(1-x)}Al_2Si_2O_8 + 2H_2CO_3 + 2H_2O = xCa^{2+} + (1-x)Mg^{2+} + 2HCO_3^- + 2SiO_2 + 2Al(OH)_3 \tag{4}$$

$$Na_xK_{(1-x)}Al_2Si_2O_8 + H_2CO_3 + H_2O = xNa^+ + (1-x)K^+ + HCO_3^- + 3SiO_2 + Al(OH)_3 \tag{5}$$

According to the dissolution equation of silicate rock, the $Ca^{2+}$, $Mg^{2+}$, and $HCO_3^-$ produced by the dissolution of silicate rock by carbonic acid are calculated according to one-fifth and one-half of the molar concentration of $SiO_4$, respectively [11]. The contributions of silicate weathering to $Ca^{2+}$, $Mg^{2+}$, and $HCO_3^-$ in the Pingzhai Reservoir basin ranged from 0.97% to 2.79% and 2.43% to 6.98%, with average values of 1.23% and 3.07%, respectively, accounting for a small proportion. This indicates that the weathering of carbonate rocks is the main process controlling the material composition and geochemical cycle of water bodies in the basin.

To show that nitric acid is indeed involved in the weathering of carbonate rocks in the basin, the ratio relationship between $[Ca^{2+} + Mg^{2+}]/[HCO_3^-]$ and $\delta^{13}C_{DIC}$ was established based on the ion and carbon isotope data in water samples. According to Figure 5, most of the sampled data fall between the endmembers of $H_2CO_3$ and $HNO_3$ dissolved carbonate rocks. This shows that $H_2CO_3$ and $HNO_3$ are jointly involved in the dissolution of carbonate rocks in the basin and have an impact on the value of carbon and nitrogen elements and ion concentration in the water body [51].

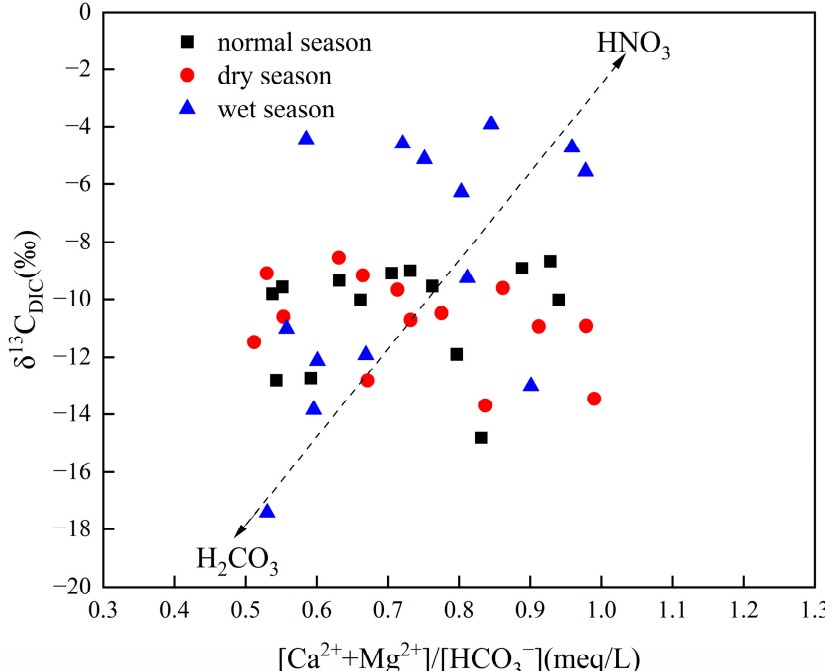

**Figure 5.** The relationship between $[Ca^{2+} + Mg^{2+}]/[HCO_3^-]$ and $\delta^{13}C_{DIC}$ in different seasons.

In addition to carbonate rock dissolution by carbonic acid, nitric acid produced by nitrification in agricultural activities also participates in carbonate rock dissolution. Assuming that carbonic acid and nitric acid participate in the dissolution process with equal molar ratios, the C–N coupling formula of carbonate rock dissolution can be obtained from Equation (6):

$$(a+b)Ca_xMg_{(1-x)}CO_3 + aH_2CO_3 + bHNO_3 = (a+b)xCa^{2+} + (a+b)(1-x)Mg^{2+} + bNO_3^- + (2a+b)HCO_3^- \quad (6)$$

where a and b represent the coefficients of $H_2CO_3$ and $HNO_3$ involved in the dissolution of carbonate rocks, respectively.

According to Equation (6), the molar ratio of $(Ca^{2+} + Mg^{2+})/HCO_3^-$ in water should be 2/3 (0.67), and the molar ratio of $(Ca^{2+} + Mg^{2+})/HCO_3^-$ in the water sample of the Pingzhai Reservoir basin was 0.17–0.98, with an average of 0.60. This indicates that the dissolution of carbonate rocks in the study area is controlled by C–N coupling, but $H_2CO_3$ and $HNO_3$ dissolve carbonate rocks according to the molar ratio of 1:1. Therefore, the amount of $(Ca^{2+} + Mg^{2+})$ and $HCO_3^-$ released by $HNO_3$ dissolved carbonate rocks can

be calculated according to the C–N coupling equation. The calculation results show that the proportion of $HCO_3^-$ produced by $HNO_3$ dissolving carbonate rocks ranges from 1.50% to 13.35%, with an average of 8.38% (Table 2). The proportion of $(Ca^{2+} + Mg^{2+})$ ranges from 4.04% to 12.06%, with an average of 7.84%. Among the different periods, the proportion of $HCO_3^-$ and $(Ca^{2+} + Mg^{2+})$ produced by $HNO_3$ dissolution during the wet season is the highest, with average values of 10.32% and 9.90%, respectively. The proportion in the dry season is the lowest, with averages of 7.29% and 5.21%, respectively. The average values in the normal season are 7.54% and 8.42%, respectively. This is because dam construction slows down or even stops the water flow, and the abundant water volume and appropriate temperature in the wet season make the water-rock interaction more complete. In addition, the amount of $NO_3^-$ fertilizer from agricultural activities increased during the wet season, which combined with $H^+$ to generate $HNO_3$ to participate in the dissolution of carbonate rocks.

**Table 2.** The ratio of $HCO_3^-$ and $(Ca^{2+} + Mg^{2+})$ produced by $HNO_3$ dissolution of carbonate rocks and the measured and theoretical values of $\delta^{13}C_{DIC}$.

| Indexes | | $HCO_3^-$-$HNO_3$ (%) | $(Ca^{2+} + Mg^{2+})$-$HNO_3$ (%) | $\delta^{13}C_{DIC}$% (Measured Value) | $\delta^{13}C_{DIC}$% (Theoretical Value) |
|---|---|---|---|---|---|
| Normal season | River | 5.82 ± 0.5 | 7.78 ± 0.4 | −12.31 ± 0.4 | −13.46 ± 0.2 |
| | Reservoir | 8.64 ± 0.3 | 8.87 ± 0.2 | −9.77 ± 0.2 | −13.06 ± 0.1 |
| | Dam | 7.31 | 8.02 | −11.48 | −13.25 |
| Dry season | River | 6.00 ± 0.7 | 5.02 ± 0.6 | −10.95 ± 0.5 | −13.43 ± 0.4 |
| | Reservoir | 7.85 ± 0.5 | 5.26 ± 0.5 | −9.85 ± 0.3 | −13.17 ± 0.1 |
| | Dam | 9.18 | 5.83 | −12.74 | −12.98 |
| Wet season | River | 9.66 ± 1.3 | 9.16 ± 0.5 | −12.73 ± 0.5 | −12.92 ± 0.3 |
| | Reservoir | 10.89 ± 0.6 | 10.88 ± 0.3 | −5.32 ± 0.3 | −12.75 ± 0.2 |
| | Dam | 9.03 | 5.74 | −13.02 | −13.01 |

Note: mean ± standard deviation (SD).

The $\delta^{13}C_{DIC}$ produced by carbonate rocks dissolved by carbonic acid is approximately −14%, and the $HCO_3^-$ produced by carbonic acid-dissolved silicate rocks is all from soil $CO_2$, so the $\delta^{13}C_{DIC}$ is approximately −23%. Carbonate rocks dissolved by nitric acid do not consume soil or atmospheric $CO_2$, and all the $HCO_3^-$ produced comes from carbonate rocks, so the $\delta^{13}C_{DIC}$ is approximately −0% [23,40,55]. The theoretical value of $\delta^{13}C_{DIC}$ in the Pingzhai Reservoir basin can be estimated by Equation (7):

$$\delta^{13}C_{DIC-T} = f_{cc}\delta^{13}C_{cc} + f_{cs}\delta^{13}C_{cs} + f_{nc}\delta^{13}C_{nc} \tag{7}$$

where $\delta^{13}C_{DIC-T}$ represents the theoretical value of $\delta^{13}C_{DIC}$ in water; $f_{cc}, f_{cs}$, and $f_{nc}$ represent the contribution proportions of carbonate rock dissolved by carbonic acid, silicate rock dissolved by carbonic acid, and carbonate rock dissolved by nitric acid to $HCO_3^-$ in water, respectively; $\delta^{13}C_{cc}, \delta^{13}C_{cs}$, and $\delta^{13}C_{nc}$ represent the values of $\delta^{13}C_{DIC}$ generated by carbonate rock dissolved by carbonic acid, silicate rock dissolved by carbonic acid, and carbonate rock dissolved by nitric acid, respectively. The calculated results show that $\delta^{13}C_{DIC-T}$ ranges from −14.06% to −12.72%, with an average value of −13.10%. The measured $\delta^{13}C_{DIC}$ ranges from −3.13% to −18.42%, with an average value of −9.32%. The measured $\delta^{13}C_{DIC}$ values in river and reservoir areas are more positive than the theoretical values in different periods, and the reservoir area is the most positive during the wet season. This is because the water flow in the reservoir area is slow and receives more light, and the photosynthesis of aquatic organisms absorbs DIC and causes isotope fractionation, making the water body $\delta^{13}C_{DIC}$ value positive [56]. Compared with the river, the proportion of $HCO_3^-$ and $(Ca^{2+} + Mg^{2+})$ produced by the dissolution of carbonate rocks by nitric acid in the reservoir water is larger, which reflects that dam construction promotes the water-rock interaction and the retention of ionic substances.

As mentioned above, carbonate rock dissolved by carbonic acid in karst systems consumes atmospheric/soil $CO_2$ to form $HCO_3^-$, one part of which is used by aquatic organisms [26,57], and the other part enters the ocean with rivers for sedimentation. In addition to carbonic acid, nitric acid is also involved in the dissolution of carbonate rocks

in the Pingzhai Reservoir. Nitrogen from chemical fertilizers (ammonium fertilizer and nitrate fertilizer), soil organic nitrogen, sewage, and livestock manure is lost and converted to $HNO_3$ (Figure 6). As a result, the concentration of $HNO_3$ in the water increases, and the carbonate rocks are dissolved by both carbonic and nitric acids. In this study, the average proportions of $HCO_3^-$ and $(Ca^{2+} + Mg^{2+})$ produced by $HNO_3$-dissolved carbonate rocks were 8.38% and 7.84%, respectively. However, this part of $HCO_3^-$ does not come from atmospheric/soil $CO_2$ but rather from carbonate rocks and does not constitute a carbon sink. Therefore, the environmental effect of C–N coupling not only causes water nitrate pollution but also reduces carbon sinks.

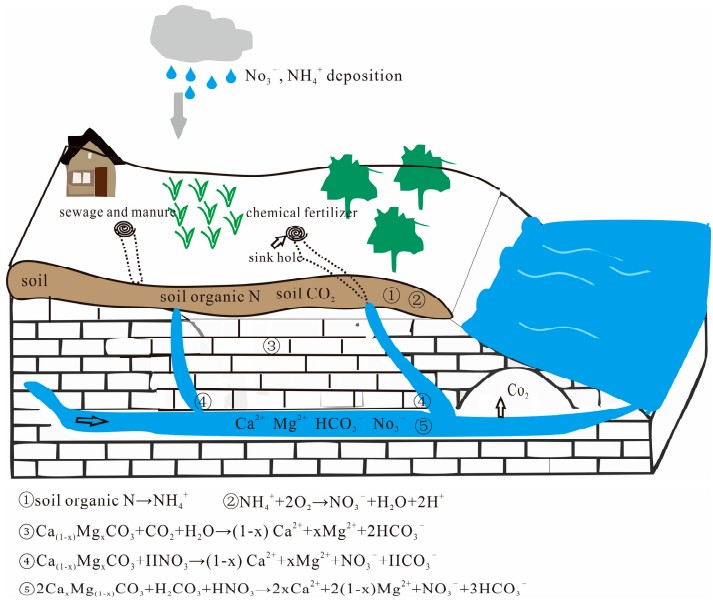

①soil organic N→$NH_4^+$　　　②$NH_4^+ + 2O_2 \rightarrow NO_3^- + H_2O + 2H^+$

③$Ca_{(1-x)}Mg_xCO_3 + CO_2 + H_2O \rightarrow (1-x)\,Ca^{2+} + xMg^{2+} + 2HCO_3^-$

④$Ca_{(1-x)}Mg_xCO_3 + HNO_3 \rightarrow (1-x)\,Ca^{2+} + xMg^{2+} + NO_3^- + HCO_3^-$

⑤$2Ca_xMg_{(1-x)}CO_3 + H_2CO_3 + HNO_3 \rightarrow 2xCa^{2+} + 2(1-x)Mg^{2+} + NO_3^- + 3HCO_3^-$

**Figure 6.** The C-N coupling cycle in the karst zone (adapted from Hu et al., 2017 [58]).

## 5. Conclusions

By monitoring the hydrochemistry and $\delta^{13}C_{DIC}$, $\delta^{15}N$-$NO_3^-$, and $\delta^{18}O$-$NO_3^-$ in the Pingzhai Reservoir and its inflow rivers, we analyzed the temporal and spatial variation in water chemistry and carbon and nitrogen isotopes, explored the source of carbon and nitrogen elements in water, and analyzed the C–N coupling in water. The results show that the dominant cation in the water of the Pingzhai Reservoir is $Ca^{2+}$, which accounts for 68% of the total cations. The dominant anion is $HCO_3^-$, accounting for 65% of the total anions. The hydrochemical type was $HCO_3$-Ca. Dissolved inorganic carbon, nitrate, and their isotopes have different spatial and temporal variations. From the river to the reservoir area and then to the outlet, the concentration of $HCO_3^-$ increased first, then decreased, and then increased again, while the $\delta^{13}C_{DIC}$ value was negative first, then positive, and then negative again. The values of $NO_3^-$, $\delta^{15}N$-$NO_3^-$, and $\delta^{18}O$-$NO_3^-$ were different in each stage of the river, which were mainly affected by dam construction and water storage, surrounding land-use mode, crop cultivation and fertilization, domestic sewage discharge, and other factors. According to the characteristics of carbon and nitrogen isotopes, the $HCO_3^-$ in the water of the study area is mainly derived from the weathering of carbonate rocks and the oxidative decomposition of organic matter. Nitrate mainly comes from agricultural activities, including chemical fertilizer (ammonium fertilizer and nitrate fertilizer), soil organic nitrogen, sewage, and livestock manure in the normal season, dry season, and wet season. The input of nitrate caused the C–N coupling cycle of hydrogeochemistry in the Pingzhai Reservoir basin and disturbed the water-rock interaction. The average proportions of $HCO_3^-$ and $(Ca^{2+} + Mg^{2+})$ produced by $HNO_3$ dissolved carbonate rocks were 8.38% and 7.84%, respectively, but this part does not constitute a carbon sink. The proportion of $HCO_3^-$ and $(Ca^{2+} + Mg^{2+})$ produced by the dissolution of carbonate rocks by $HNO_3$ in

reservoir water was relatively large. This reflects the full water-rock interaction and retention effect due to the construction of the dam. Therefore, successful fertilizer application experience and farmland management practice should be learned, controlling nitrogen input from agricultural activities. Using new technologies can increase the capacity of domestic wastewater treatment and limit the discharge of sewage into rivers and reservoirs and prevents soil organic nitrogen loss. In addition, a limitation of the study is that it is lacking an analysis of the impact of geological conditions on the geochemical characteristics of carbon and nitrogen elements. This will be addressed in subsequent studies.

**Author Contributions:** Z.Z.: Resources, Data curation, Formal analysis, Project administration, Funding acquisition. J.K.: Conceptualization, Methodology, Software, Writing—original draft. F.Z.: Visualization. Y.Z.: Supervision. J.X.: Investigation. C.W.: Validation. All authors have read and agreed to the published version of the manuscript.

**Funding:** National Natural Science Foundation of China (42161048); National Natural Science Foundation of China (41661088); Science and Technology Plan Project of Guizhou Province (Qiankehe Jichu [2020]1Y154) financially supported this study.

**Data Availability Statement:** Not applicable.

**Acknowledgments:** This study was supported financially by the State's Key Project of Research.

**Conflicts of Interest:** The authors declare no conflict of interest.

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
