# Peer review of "Study on the Carbon and Nitrogen Isotope Characteristics and Sources and Their Influence on Carbon Sinks in Karst Reservoirs"

_land, doi:10.3390/land12020429_

Round 1

Reviewer 1 Report (Previous Reviewer 2)

I reviewed this article last month,The author has carefully revised and answered my questions, and I think it is ready for publication

Author Response

Reviewer 2 Report (Previous Reviewer 4)

The MS has adequately addressed all my comments and can be accepted in the present form.

Author Response

Reviewer 3 Report (New Reviewer)

The manuscript entitled: "Study on the carbon and nitrogen isotope characteristics and sources and their influence on carbon sinks in karst reservoirs" is an interesting study and, in my opinion, is worth to be published. The authors provided new interesting results. The structure of the manuscript is well prepared. However, I wouldn't say I like the manuscript situation in which the crucial results appeared only in the discussion to support the findings and the discussion flow. On the other hand, those results are suitable and present in the discussion section, which does not disturb the manuscript's readability. If this structure is acceptable in the Land journal, I have only a minor suggestion which the authors might find in the attached pdf file with my comments.

Author Response

This manuscript is a resubmission of an earlier submission. The following is a list of the peer review reports and author responses from that submission.

Round 1

Reviewer 1 Report

REVIEW COMMENTS

Study on carbon and nitrogen isotope characteristics, source and its influence on carbon sink in karst reservoirs

Zhongfa Zhou et al.

General Comments

The manuscript of Zhou et al. presented how carbon and nitrogen isotope values fluctuate depending on the conditions of the study area. The main methodology used in this research is hydrochemical analyses. This manuscript also discusses the source of the value fluctuations and the consequences to pollution in karst reservoirs. Aside from presenting the different values and noting the fluctuations depending on the season and proximity to the reservoir and/or river, they also pinpoint the possible anthropogenic activities (particularly agriculture-related) that caused the observed values. The authors argue that the values observed in this study reflect the water-rock interaction in the study area as well as the effect of the dam construction.

Yet, the paper lacks seriously in following points:

1.     Poor English use and frequent grammatical error make it difficult to understand and follow the flow of manuscript. This manuscript should not be accepted without English proofreading by native speaker as almost every sentence contains grammatical error. I included few examples below.

2.     Many of the sentences in this manuscript are too long that the authors end up failing to convey what they want to say clearly. Most of the sentences are also poorly constructed and can use some revising. Many concise sentences are better than sentences that are too long and confusing.

3.     The use of Oxford comma is recommended since the manuscript often uses enumerations or list of examples, so the lack of the extra comma before the word “and” makes some concepts confusing.

4.     Unnecessary capitalization is found in many words in the manuscript.

5.     The Abstract of the manuscript exceeded the 200-word count limit. Please revise the abstract accordingly.

6.     The limitations of the study are not clearly described anywhere in the paper.

7.     Since the methodology is very detailed, it might be helpful to include a flowchart or make a supplementary paper/annex for this part.

8.     Literature review in the Introduction part is very minimal and the concepts are not organized well.

9.     The Results and Discussion sections mentions ranges of values. Please check if the proper term used should be “ranges from ___ to ___” instead of “ranges were” when talking about the range of values.

10.  No further recommendations were mentioned at the conclusions section.

Specific Comments

This paper lacks in many aspects that it falls short to meet the high standard of our journal.

1.     Lines 16-19 on p. 1: The result which uses the words “increased first, then decreased…” can be better summarized and revised accordingly.

2.     Lines 22-23 on p. 1: Notice how the lack of Oxford comma in the phrase “normal, dry and wet seasons” makes the statement confusing. Please consider adding an extra comma after the word “dry” so that the readers are aware that we are talking about three different seasons instead of 2 normal seasons.

3.     Lines 34 on p. 1: Why is the “C” in “Connect” capitalized?

4.     Lines 35-39 on p. 1: This sentence is too long, please consider separating into shorter sentences.

5.     Lines 40-43 on p. 1: This sentence should be rephrased/rearranged to clarify the idea.

6.     Lines 43-44 on p. 1: Please consider the use of an Oxford comma in this situation.

7.     Line 46 on p. 2: Why is the “B” in “Biogeochemistry” capitalized?

8.     Lines 47-84 on p. 2: This paragraph is too long, please consider splitting into two or more properly grouped paragraphs.

9.     Lines 70-74 on p. 2: The sentence is long but seems to lack some necessary words to make the thought of the sentence complete. Please consider reorganizing and splitting into multiple sentences.

10.  Line 86 on p. 2: It is suggested to change “and the exposed area of…” to “with an exposed area of…” or something to that effect.

11.  Line 87 on p. 2: What do you mean by “hot area”? Kindly considering using a more appropriate term.

12.  Lines 96-104 on pp. 2-3: Please consider restating the objectives. The numbering did not help clarify the specific objectives of the paper. Instead of using numbers to explicitly enumerate the different objectives, it might be better to reorganize the sentences and avoid using sentences that are too lengthy so that the statement of objectives is clearer and more concise.

13.  Line 109-110 on p. 3: Please consider the use of Oxford comma in this sentence.

14.  Lines 114-116 on p. 3: The description of the dam uses sentences that are too long. Please consider re-organizing this section and maybe including an appropriately annotated photo of the dam so that the readers can be able to appreciate the figures being mentioned in this statement.

15.  Lines 124-125 on p. 3: The word “which belongs to” does not seem appropriate for this sentence. Please consider revising.

16.  Lines 127-129 on p. 3: It maybe helpful to cite Figure 1 in the text. Also, because it is better to mention the figure in the text first before showing the figure itself.

17.  Lines 137-143 on p. 3: The author presents valid discussion points in this section but the whole paragraph lacks organization. It would be better to rewrite the sentences to fit what the author is trying to convey.

18.  Line 148 on p. 4: “According to” can be changed to “In accordance with”.

19.  Lines 165-166 on p. 4: The tone of this last sentence is not consistent with the rest of the paragraph. Did you refrigerate the samples for your study or is this part just a recommendation? Please consider revising this sentence accordingly.

20.  Line 171 on p. 4: Inductively Coupled Plasma-Emission Spectrometer must be capitalized accordingly.

21.  Line 175 on p. 4: The term “blow in” can be replaced with a more appropriate verb.

22.  Lines 256-257 on p. 7: Please use the actual terms instead of putting “>” symbols.

23.  Line 266 on p. 7: Before using the acronym DIC, mention or define it first as dissolved inorganic carbon during its first appearance in the text (line 165 on p. 4).

24.  Line 284 on p. 7: The statement “a trend of slowly increased and then decreased” is very ambiguous and grammatically incorrect.

25.  Lines 290-295 on p. 7: Please double check the format when enumerating your reasons.

26.  Line 316 on p. 8: The use of the term “which reflected” is very confusing. Perhaps it should be changed to “which is a consequence of” or another similar term.

27.  Lines 350-351 on p. 9: The sentence “Using δ15N-NO3- and δ18O-NO3- can trace source of nitrate.” is vague and incomplete. Please consider adding a follow-up sentence or explaining the context in the same sentence.

28.  Lines 376-378 on p. 9: This sentence is poorly constructed. Please consider revising this.

29.  Lines 387-390 on p. 10: This sentence is too long, making the statement confusing.

30.  Line 402 on p. 10: Why is the “M” in “Marine” capitalized?

31.  Line 403 on p. 10: Why is the “K” in “Karst” capitalized?

32.  Line 413 on p. 10: Please change “this” to “which”.

33.  Line 414 on p. 10: Please change “contributing” to “which contributes”.

34.  Line 427 on p. 11: The word “calculate” should be in past tense.

35.  Line 446 on p. 11: Omit “As mentioned above,” because it is unnecessary.

36.  Line 459 on p. 12: Please revise the phrase “dissolve carbonate rocks do not according” because it is confusing and grammatically incorrect.

37.  Line 470 in p. 12: What do you mean by “more fully”? Kindly consider rephrasing this sentence.

38.  Line 491 on p. 12: Please replace “due to” with “because”.

39.  Lines 491-493 on p.12: The sentence is confusing, please consider revising and correcting grammatical errors to clarify the meaning.

40.  Lines 495-496 on p. 12: The part “which reflects that due to dam construction makes the water-rock interaction is fully” is both grammatically incorrect and confusing. Please revise.

41.  Lines 515-520 on p. 13: The first sentence of the Conclusions section is too long. Please split this into two or more sentences.

42.  Lines 524-526 on p. 13: Please consider a better way of stating the fluctuations in the values.

Reviewer 2 Report

Comments:

This is a meaningful manuscript. Much of the basic work is original and could potentially contribute to understanding the mechanism of karst process related carbon sink impacted by carbon  nitrogen coupling effect in carbonate rock distribution territory. The manuscript is a case study of hydrogeochemical characteristics of carbon and nitrogen elements and stable isotopes in humid subtropical karst river and reservoir continuum. However, as far as the international journal is concerned, a more in-depth analysis of the original data and improvement of scientific understanding of the findings are needed as a case study manuscript. I recommend a major revision before it can be accepted for publication.

1) According to the title, the manuscript mainly focuses on the carbon sink of reservoirs in karst areas. However, the study content only focuses on a reservoir located in karst hilly area of Southwestern China. Therefore, it is necessary to analyze the representativeness and typicality of the Pingzhai reservoir in order to understand the overall regularity through typicality.

2) There is an obvious tendency in the manuscript to weaken the geological and hydrogeological background of the study area, which plays an important role in controlling the hydrogeochemical characteristics. The lithology of strata flowing through rivers in the study area is different, so it is suggested to analyze the impact of geological conditions on the geochemical characteristics of carbon and nitrogen elements in a new separate section.

3) In Figure 1, only the chronostratigraphic map is shown, but there is no geomorphological information, which is not conducive to further understanding of the basic conditions of the reservoir basin. It is suggested to supplement geomorphological and land use maps of the study area. In addition, the basin area parameters of Pingzhai Reservoir are supplemented in the proper position in the paper.

4) Mixing is one of the important roles in the formation of hydrochemical elements and stable isotopes, and water flux are crucial to the mixing calculation. Pingzhai Reservoir is formed by the recharge of five rivers (NYR, SGR, ZWR, BSR, HJR), and the runoff flux is different. It is suggested to supplement some hydrological parameters when analyzing the variation law of carbon and nitrogen elements in reservoir water.

5) Line 442: There is a mistake in the abscissa unit in Figure 5.

6) Error analysis should be made between measured and theoretical values of stable carbon isotope of DIC in Table 2.

7) Further summarize the contents from line 501 to line 513, and supplement the related carbon-nitrogen coupling cycle mode diagram.

8) This work could benefit from review and polish by native English speaker.

Reviewer 4 Report

General comments:
This paper through the sampling analysis and systematic monitoring of water chemistry, dissolved inorganic carbon isotopes, nitrogen and oxygen isotopes of reservoirs and their inflow rivers in karst area, the source and spatial-temporal variation characteristics of carbon and nitrogen elements in water were analyzed, and the C-N coupling cycle and its influence on karst carbon sink were discussed in the Pingzhai karst plateau reservoir. This is a timely research theme, and of significance for understanding the effects of nutrients (specifically nitrate) within karst on dissolution of carbonate rock. The vulnerability of karst environments to pollution dynamics adds urgency and impact to this broad research theme. The research is, on the whole, well executed and a significant amount of time and effort has clearly been expended on data collection and analysis, besides, the article is potentially useful and within the research interests of the journal. However, the current format of the paper and the writing style is difficult to follow in places and there is a lack of clarity in key messaging some section. Some sections descend into a level of detail which lacks relevance to the science presented, and other sections are needs to be revised with regards to data interpretation. Key parts of the methodology require refining. There are also many references omitted. The study of nitrate in karst is a limited field, but there has not been an exhaustive search of all available literature. Moreover, the novelty of the paper should be highlighted. I have placed detailed review comments.

Overall, this is a good piece of work which will have good impact in the field, if the key findings can be communicated in a clearer way and better credit to existing literature in the field can be provided.

The following points need to be modified:

Abstract

(1) The abstract should be written in the order of research purpose, research method and research results, and the results should be combed and condensed.

Introduction

(2) Water bodies in karst areas are generally rich in calcium and somewhat alkaline. (Reference is needed here)

(3) The authors focus on some case studies. My expectation would be to discuss similar case studies from karstic areas worldwide given that “karst” appears in the title.

(4) The objectives did not include the biogeochemical processes, which is critical when using nitrate isotope techniques. I think the objectives are a bit weak, without innovation and not set in an international context that could attract the reader.

Materials and methods

(5) Figure 1: Please fix the legend. It should be “Reservoir samples”.

(6) The test accuracy of δ15N-NO3- and δ18O-NO3- was 0.3‰. (Need quote to 1SD)

Results

(7) 3.2. Characteristics of DIC and δ13CDIC in Pingzhai Reservoir

 “Due to the influence of the dam ….. in the surface water”. This part is confusing. How does the value of δ13CDIC change? Please re-write to improve clarity.

Discussions

(8) 4.1. Spatial variation and influencing factors of dissolved inorganic carbon and nitrate

“The δ13CDIC value of water body ….. and increase of DIC concentration”.

Can you be more specific? Which water body? Whether or not continuing the theme of the previous sentences?

(9) 4.1. Spatial variation and influencing factors of dissolved inorganic carbon and nitrate

“In the reservoir area, the water was fully mixed, and the nitrate concentration has little difference among the sampling points in different seasons”.

How the fully mixed is justified?

(10) 4.2. Sources of dissolved inorganic carbon and nitrate

“Using δ15N-NO3- and δ18O-NO3- can trace the source of nitrate”.

“Nitrate from different sources has different δ15N and δ18O values”.

“Combined with the δ15N-NO3- and δ18O-NO3- value of water body in the study area, various nitrate sources could be determined.”

This part is redundant, please delete it or move it to introduction for brevity.

(11) 4.2. Sources of dissolved inorganic carbon and nitrate

“This part of the lost N will be converted into nitric acid through nitrification and will participate in the geochemical cycle,”

This is confusing. How nitric acid is formed? The authors should re-write it.

(12) 4.3. Weathering of carbonate rocks and C-N coupling relationship in Pingzhai reservoir basin

“In addition, the [Ca2++Mg2+]/[HCO3-] equivalent ratios in the water samples in the study area were all greater than 1 (1.37 on average),”

The author claims that [Ca2++Mg2+]/[HCO3-] equivalent ratios are greater than 1, and Ca2+and Mg2+have additional sources. But they need to provide some evidence.

(13) Organize the discussions section and check for missing references.

Conclusions

(14) The authors just repeat the results and looks like the abstract. The main findings and some future directions should be summarized instead.

I hope these comments help in refining the work.

With best wishes.